# 3′-*O*-β-d-glucopyranosyl-α,4,2′,4′,6′-pentahydroxy-dihydrochalcone, from Bark of *Eysenhardtia polystachya* Prevents Diabetic Nephropathy via Inhibiting Protein Glycation in STZ-Nicotinamide Induced Diabetic Mice

**DOI:** 10.3390/molecules24071214

**Published:** 2019-03-28

**Authors:** Rosa Martha Pérez Gutierrez, Abraham Heriberto García Campoy, Silvia Patricia Paredes Carrera, Alethia Muñiz Ramirez, José Maria Mota Flores, Sergio Odin Flores Valle

**Affiliations:** 1Natural Products Research Laboratory, Higher School of Chemical Engineering and Extractive Industries, National Polytechnic Institute, Av. Instituto Politécnico Nacional S/N, Unidad Profesional Adolfo Lopez Mateos, Ciudad de México CP 07708, Mexico; abrahamhgc27@hotmail.com (A.H.G.C.); josemariamota@yahoo.com.mx (J.M.M.F.); 2Sustainable Nanomaterials Laboratory, Higher School of Chemical Engineering and Extractive Industries, National Polytechnic Institute (IPN) Professional Unit Adolfo Lopez Mateos, S/N Av. Instituto Politécnico Nacional, Ciudad de México CP 07708, Mexico; silviappcar@gmail.com; 3CONACYT/IPICYT-CIIDZA, Camino a la Presa de San José 2055, Col. Lomas 4 Sección, San Luis Potosí CP 78216, Mexico; alethiamura@gmail.com; 4Green Chemistry Research Laboratory, School of Chemical Engineering and Extractive Industries, National Polytechnic Institute, Av. Instituto Politécnico Nacional S/N, Unidad Profesional Adolfo Lopez Mateos, Ciudad de México CP 07708, Mexico; sergioodin@gmail.com

**Keywords:** dihydrochalcone, advanced glycation end-product, *Eysenhardtia polystachya*, diabetic mice, renoprotective

## Abstract

Previous studies have shown that accumulation of advanced glycation end products (AGEs) can be the cause of diabetic nephropathy (DN) in diabetic patients. Dihydrochalcone 3′-*O*-β-d-glucopyranosyl α,4,2′,4′,6′-pentahydroxy–dihydrochalcone (**1**) is a powerful antiglycation compound previously isolated from *Eysenhardtia polystachya*. The aim was to investigate whether (**1**) was able to protect against diabetic nephropathy in streptozotocin (STZ)-induced diabetic mice, which displayed renal dysfunction markers such as body weight, creatinine, uric acid, serum urea, total urinary protein, and urea nitrogen in the blood (BUN). In addition, pathological changes were evaluated including glycated hemoglobin (HbA1c), advanced glycation end products (AGEs) in the kidney, as well as in circulation level and pro-inflammatory markers ICAM-1 levels in diabetic mice. After 5 weeks, these elevated markers of dihydrochalcone treatment (25, 50 and 100 mg/kg) were significantly (*p* < 0.05) attenuated. In addition, they ameliorate the indices of renal inflammation as indicated by ICAM-1 markers. The kidney and circulatory AGEs levels in diabetic mice were significantly (*p* < 0.05) attenuated by (**1**) treatment. Histological analysis of kidney tissues showed an important recovery in its structure compared with the diabetic group. It was found that the compound (**1**) attenuated the renal damage in diabetic mice by inhibiting AGEs formation.

## 1. Introduction

Type 2 diabetes mellitus is a cause of mortality due to complications such as diabetic nephropathy (DN), which is the main cause of end-stage renal failure [1]. DN is characterized by kidney structural changes, declining glomerular filtration rate, and mesangial sclerosis. Hyperglycemia, oxidative stress, and dyslipidemia are the main causes of increased advanced glycation end products (AGEs) and contribute to the development of DN in diabetes [2], which has been demonstrated in several studies with the treatment of anti-glycation compounds such as alagebrium chloride [3], benfotiamine [4], pyridoxamine [5], and aminoguanidine [6] attenuated development of diabetic nephropathy. The generation of AGEs promotes kidney damage by protein cross linking, leading to changes in structure and function of the proteins [7]; they also generate an increase of the expression of monocyte chemoattractant protein 1 (MCP-1). Previous studies indicated that endothelial cell exhibition to a uremic environment augment IL-8, vascular adhesion molecule-1 (VCAM-1) and MCP-1 expression indicating a relationship between systemic inflammation and vascular damage with a uremic toxicity [8]. Other studies indicated that reduction of AGEs production and accumulation in the tissues could be an effective strategy to improvement of diabetic complications [9].

In previous studies, we isolated several dihydrochalcones from the Bark of *Eysenhardtia polystachya* [10], which showed an efficient inhibition fluorescent and non-fluorescent AGE formation, reduced level of fructosamine, significantly suppressed oxidation of thiols and protein carbonyl content in a BSA/glucose system; in addition, inhibited generation of MGO, and the formation of amyloid cross-β structure. Dihydrochalcone demonstrated inhibition at multiple stages of glycation. The aim was to isolate a dihydrochalcone and study if it can be renoprotective in diabetic mice by inhibiting AGEs formation.

## 2. Results and Discussion

### 2.1. Identification of 3′-O-β-d-glucopyranosyl α,4,2′,4′,6′-pentahydroxy–dihydrochalcone (**1**)

Dihydrochalcone was first found in the methanol extract of *Eysenhardtia polystachya*, and its structure was elucidated by spectroscopic methods (IR, ^1^H-NMR, ^13^C-NMR, COSY, and HMBC). Compound **1** shows a molecular formula C_21_H_24_O_12_, and this is suggested by using the positive HRMS that implied ten degrees of unsaturation. The IR spectrum indicates the presence of hydroxyl at 3448 cm^−1^, carbonyl (1645 cm^−1^) and benzene ring (3010, 1586, 1441 cm^−1^) functionalities.

^1^H, ^13^C-NMR, and DEPT spectra show the presence of five aromatic signals at δH 7.49, δH 6.65, and δH 6.39, and also showed signals for 21 C-atoms, including two methylene (CH_2_), eleven methylenes (CH) groups, and eight quaternary carbons. In the ^1^H-NMR spectrum there are five aromatic signals at δH 7.49 (2H), δH 6.65 (2H) and δH 6.09 (1H). In the HMBC spectrum correlation of δH 7.49 with δC 131.6 (C-2, 6), 6.65 with δC 118.2 (C-3, C-5) suggested the presence of one monosubstituted aromatic ring with a hydroxyl group at C-4 (δC 160.7). The glucopyranosyl moiety was located at C-3′ which was further supported by the HMBC correlation of the anomeric proton resonating at δ_H_ 4.80 (1H, d, *J* = 7.3 Hz, H-1″) to C-3′ (102.6) together with the coupling constant of the anomeric proton (d, *J* = 7.3 Hz) indicated that was β-glucoside.

2′,4′,6′-trihydroxy substitution of the dihydrochalcone ring A was determined by evaluating the ^1^H coupling pattern and the ^13^C-NMR chemical shifts which showed the characteristic pattern of nethofagin dehydrochalcone skeleton [11] except that (**1**) differs from nethofagin by the presence of a hydroxyl group at C-8 (C-α). Aliphatic proton was assigned to ABX system at δ_H_ 2.75 (1H, dd, *J* = 16.5, 13.4 Hz, H-α), 2.36 (1H, dd, *J* = 16.5, 3.1 Hz, H-α), 5.25 (1H, dd, *J* = 13.4, 3.1 Hz, H-β), suggested a linked -CH (α)-CH_2(_β)- moiety leading to the presence of a -CO-CH (OH)-CH_2_- moiety. Therefore, compound **1** was assigned as 3′-*O*-β-d-glucopyranosyl-α,4,2′,4′,6′-pentahydroxy-dihydrochalcone (Figure 1).

### 2.2. Effect of Dihydrochalcone on Glucose, Water Intake, Body Weight, Kidney Weight, Food Consumption, Urine Volumen and Urine Protein

As shown in the Figure 2 urine volume (A), food consumptions (B), water intake (C), and body weight (D) were significantly higher when compared to that of the normal control mice. Treatment with compound (**1**) at a dose of 100 mg/kg during a period of 5 weeks significantly decrease the urine volume by 52.68% (Figure 2A; *p* < 0.05) and urine protein by 52.22% compared to that diabetic control group (Table 1). While treatment with (**1**) food consumptions and water intake decreased by 32.75% (Figure 2B; *p* < 0.05) and 53.84% (Figure 1C; *p* < 0.05), respectively, compared to that diabetic control group. However, using compound (**1**) during the 5 weeks of treatment did not significantly modify levels of blood glucose in STZ-induced diabetes mice, which developed a stable increase in the hyperglycemia. In addition, treatment with compound (**1**) or metformin did not show significant changes in body weight (Figure 2D). During the period of experimental treatment with compound (**1**), there was an improvement in urine volume, urine protein, food consumption, and water intake in different degrees when compared to metformin, which was used as standard (Figure 2A–D). Figure 3A shows kidney size during the experimental period; it was observed that a gain in kidney size in the diabetic control mice was in contrast to a reduction of kidney size when the diabetic mice were treated with compound (**1**) or metformin for 5 weeks.

Kidney weight was significantly increased (*p* < 0.05) in the diabetic group compared to the control group, while oral administration with compound **1** (100 mg/kg) and metformin (200 mg/kg) exhibited a significant reduction (*p* < 0.05) in kidney weight by 32% and 28.6%, respectively, as compared to diabetic-STZ mice (Figure 3B).

It was observed that in the STZ-induced diabetic model that a selective destruction of pancreatic cells producing insulin led to hyperglycemia. Consequently, we have an experimental diabetic nephropathy model to study pathological changes in the kidney [12]. Hyperglycemia produces an increase in urine volume, urine protein, food consumption, water intake, blood glucose level, and reduction of body weight [13]. Treatment with compound (**1**) significantly enhances these pathological characteristics in the DN mice model. The finding indicated that the renoprotective effect of compound **1** on the DN model is related to the improvement of renal function, and avoids the proteinuria. However, its effect does not depend on changes in blood glucose levels.

### 2.3. Effect of Dihydrochalcone on Kidney Index, Creatinine, Uric Acid, Serum Urea, and Urea Nitrogen in the Blood (BUN)

The development of diabetic nephropathy can be detected by the elevated level of kidney index indicators such as uric acid, serum urea, BUN, and creatinine. These renal indexes were significantly higher in the STZ-induced diabetic mice group (DN) in comparison with normal control. Groups treated with 25, 50, and 100 mg/kg of the dihydrochalcone showed a significantly decreased (*p* < 0.05) in kidney index in comparison to DN control group such as BUN, creatinine, uric acid, urea and urine protein in a dependent dose manner (Table 1 and Table 2). In addition, metformin used as a standard also significantly reduced (*p* < 0.05) the elevated level of these biomarkers. STZ-induced diabetic mice showed a significant increase in volume and kidney mass. However, treatment with compound 1 (25, 50 and 100 mg/kg) or metformin (200 mg/kg) significantly re-established (*p* < 0.05) both volume and renal mass look closer to the normal group (Figure 3A,B).

Atrophic changes in the renal tubules and glomeruli in the kidneys were observed in the DN mice model, producing elevated levels of blood urea nitrogen, urea, uric acid, and creatinine in blood. An important index of glomerular function is the creatinine generated as a metabolite in the muscle, excreted through glomerular filtration [14]. In the metabolic process, protein breakdown leads to the production of urea, which is excreted mainly through the kidneys [15]. Metabolism in humans serum urea nitrogen level is the main end-product of proteins causing a spike in serum, BUN level is commonly found in DN or glomerulonephritis patients with inhibition glomerular filtration rate. The accumulation of uric acid generates the production of monosodium urate crystals that can cause inflammatory and pain response, leading to renal and hepatic injuries. High levels of serum urea, BUN, uric acid, and creatinine suggest injuries in the kidney [16]. Thus, it could be used as markers for diagnosis in DN [17].

### 2.4. Effect of Dihydrochalcone on MCP-1

Diabetic nephropathy is considered an inflammatory disease where progressive glomerular damage is related by infiltration to CD11b-positive macrophages [18] wherein MCP-1 is secreted from resident glomerular cells. In our study, the diabetic model is related to the increase in MCP-1 and simultaneously with accumulation of macrophages in the cortex. Treatment with compound (**1**) and metformin were able to significantly reduce (*p* < 0.05) the expression of the chemokine MCP-1 associated with diabetes (Figure 4). The finding suggested that the anti-inflammatory effect of compound (**1**) participates in DN attenuation. AGEs-induced chronic inflammation, and subsequently cells and tissues, were disabled and triggered an increasing inflammatory cytokines or oxidative stress via interaction between RAGE and AGEs [19].

### 2.5. Effect of Dihydrochalcone on Glycosylated Haemoglobin Concentration (HbA1c)

The level of glycosylated hemoglobin in STZ-induced diabetes mice was significantly increased (*p* < 0.05) in comparison with normal control mice. However, HbA1C significantly decreased (*p* < 0.05) in groups treated with compound **1** in comparison with diabetic group (Table 3) at a dose of 25, 50, and 100 mg/kg by 19.47%, 33.65% and 54.49% respectively. Metformin also significantly reduced (*p* < 0.05) the elevated concentration of glycosylated hemoglobin by 48.14% (Table 3).

In diabetes, the excess of glucose in blood reacts with hemoglobin to form HbA1C, which is an early glycosylation adduct, and with time undergoes complex and slow rearrangements to generate AGEs. Thus, glycosylated hemoglobin is used in diabetic patients mostly for prognosticate the developing of diabetic complications, mainly at long term [20].

### 2.6. Effect of Dihydrochalcone on AGEs Kidney

In diabetes hyperglycemic condition leads to excessive accumulation of AGEs participating in the pathogenesis of diabetic nephropathy, leading to structural abnormality [4]. In this study, treatment with STZ markedly increased AGEs formation in kidneys of diabetic mice in comparison to normal control. Groups treated with dihydrochalcone (100 mg/kg) or metformin (200 mg/kg) showed attenuation in AGEs formation in kidney in comparison to the diabetic control group by 47.73% and 50% respectively (Table 3). Results showed that compound (**1**) is able to block the accumulation and formation of AGEs in kidney in nephropathy model streptozotocin induced diabetic mice. Reduction of the formation of AGEs is considered a potential therapy in diabetic nephropathy.

### 2.7. AGEs Levels in Plasma

AGEs levels in plasma showed a significant increase in 5 weeks of diabetes induction. However, treatment of compound **1** to diabetic mice showed progressive decrease circulating AGEs in respect to diabetic control mice (Figure 5A,B). AF show that are better related to the time of illness than AGI. Plasma AGE levels circulation stopped incrementing, possibly because it is eliminated from blood by liver and renal filters. This study showed higher serum AGE level in diabetic mice and an increase in renal dysfunctions. Oral administration of compound (**1**) act as AGE-inhibitors near the normal levels as well as AGE-breakers, reducing the serum AGE level and delay the progression of DN. In addition, metformin treatment showed AGEs level near to normal levels compared to the diabetic group (Figure 5A,B); this effect is due to its interaction with dicarbonyl compounds formed throughout the glycation process [21].

### 2.8. Histopathology on in Renal Tissues

The renal tissue of control mice revealed normal parenchyma, renal cortex, renal tubules, and glomeruli (Figure 6A). Sections of STZ-induced diabetic mice kidney showed perivascular lymphocytic aggregates, endotheliosis, inflammatory cell infiltration, degenerative changes, fibrotic, interstitial hemorrhage, and glomerular necrosis (Figure 6B). Administration of 100 mg/kg (Figure 6E) of dihydrochalcone, or 200 mg/kg of metformin (Figure 6F) to the experimental animals showed an important recovery in the structure of the kidney compared to the diabetic group. However, treatment a 25 and 50 mg/kg showed a moderate improvement on normal glomerulus compared to the diabetic group.

## 3. Conclusions

Our findings suggest that the treatment with dihydrochalcone protects renal function and prevents kidney injury in STZ-nicotinamide induced diabetic nephropathy, ameliorated markers of DN, as well as inflammation, HbA1C, AGE-inhibition in kidneys and circulation.

The renoprotective effect of dihydrochalcone isolated from *Eysenhardtia polystachya* might be associated in part to its ability to react with reactive carbonyl species and cleavage of pre-formed AGEs within the kidney by a cross- link breaker inhibiting AGEs-formation.

## 4. Materials and Methods

### 4.1. Plant

The specimen was identified and authenticated by Biol. Aurora Chamal, Department of Botany, Universidad Autonoma Metropolitana-Xochimilco, where a voucher specimen (No. 53290) has been deposited for further reference.

### 4.2. Extraction, Isolation and Characterization 3′-O-β-d-glucopyranosyl α,4,2′,4′,6′-pentahydroxy–dihydrochalcone (**1**)

The extraction, isolation, and characterization of dihydrochalcone from the bark of *Eysenhardtia polystachya* was carried out as follows. Briefly, the bark (40 kg) was pulverized into powder and extracted with distilled water and methanol (1:1) two times at room temperature. Both extracts were combined and concentrated under reduced pressure. The extract was subjected to a silica gel column eluted with ethyl acetate/methylene chloride (2:9) to yield seven fractions (PA-1 to PA-7). Subfraction PA-5 was then separated by silica gel chromatography with methanol/acetone/ethyl acetate (1:3:6) and preparative chromatography eluted with methanol/acetone/ethyl acetate (0.5/3/1.5) to give 4 subfractions (PA5-1 to PA5-4). PA5-3 was purified in a sephadex LH-20 column with a gradient of water/methanol 1:1 increasing the ratio of water to 100% to obtain compound **1** (630 mg).

*3′-O-β-d-glucopyranosyl-α,4,2′,4′,6′-pentahydroxy–dihydrochalcone:* Is a pale yellow powder, m.p. 143–144 °C; HRMS [M^+^] at *m*/*z* 468.4130, C_21_H_24_O_12_ requires 468.4110; UV λ max (MeOH) nm 260, 303, 375; IR νmax (KBr) cm^−1^: 3448, 2932, 1645, 1586, 1519, 1441, 1182, 1071; The ^1^H-NMR espectra (300 MHz, DMSO-d6) showed the following data: δ_H_ 2.75 (1H, dd, *J* = 16.5, 13.4 Hz, H-α), 2.36 (1H, dd, *J* = 16.5, 3.1 Hz, H-α), 5.25 (1H, dd, *J* = 13.4, 3.1 Hz, H-β), 7.49 (2H, d, *J* = 8.2 Hz, H-2, 6), 6.65 (2H, d, *J* = 8.2 Hz, H-3, 5), 9.10 (s, OH-4), 6.09 (1H, s, H-5′), 10.60 (s, OH-4′), 12.24 (s, OH-2′), 13.50 (s, OH-6′); Glu: δ_H_ 4.80 (1H, d, *J* = 7.3 Hz, H-1′’), 3.87 (1H, d, *J* = 4.4 Hz, OH-2′’), 3.54 (1H, d, *J* = 5.3 Hz, OH-3′’), 3.28 (1H, dd, *J* = 12.2, 4.5 Hz, OH-4′’), 3.20 (1H, d, *J* = 9.3 Hz, OH-5′’), 3.76 (1H, dd, *J* = 11.6, 5.2 Hz, H-6″α), 3.45 (1H, dd, *J* = 11.6, 6.0 Hz, H-6″β); *^13^*C NMR (125 MHz, CDCl3) δ_C_: 128.7 (C-1), 131.6 (C-2, C-6), 118.2 (C-3, C-5), 160.7 (C-4), 44.30 (C-α), 78.97 (C-β), 192.94 (C-9, C=O); 109.4 (C-1′), 164.2 (C-2′), 102.6 (C-3′), 166.2 (C-4′), 111.5 (C-5′), 165.7 (C-6′); Glu: δ_C_ 108.6 (C-1″), 72.3 (C-2″), 78.0 (C-3″), 69.7 (C-4″), 76.8 (C-5″), 60.4 (C-6″).

### 4.3. Animals

The study was conducted on healthy adult male C57BL/6J mice, weighing about 25–30 g. Before and during the experiment, animals were fed a standard laboratory diet (Mouse Chow 5015, Purina) with free access to water. Mice were procured from the bioterium of ENCB and were housed in microloan boxes cages in a controlled environment (temperature 25 ± 2 °C). Animals were allowed to acclimate for a period of three days in their new environment prior to the study. Before commencing the experiment, litter in cages was renewed three times a week to ensure hygiene and maximum comfort for the animals. The experiments reported in this study followed the guidelines stated in “Principles of Laboratory Animal Care” (NIH publication 85-23, revised 1985 and the Mexican Official Normativity (NOM-062-Z00-1999). All animal procedures were performed in accordance with the recommendations for the care and use of laboratory animals (756/lab/ENCB).

### 4.4. Induction of Mild Diabetes (Type 2)

After 15 min of administrating an intraperitoneal injection with 120 mg/kg nicotinamide (Sigma Chemical Company, St. Louis, MO, USA), mice were made diabetic by administering a single intraperitoneal injection of freshly prepared streptozotocin (STZ) (60 mg/kg b.w. i.p.) in 0.1 M citrate buffer at pH 4.5. The animals were allowed to drink 5% glucose solution overnight to overcome the drug-induced hypoglycaemia. After 10 days of diabetes development, moderately diabetic mice having persistent glycosuria and hyperglycaemia (blood glucose > 200 mg/dL) were used for further experimentation [22]. After the diabetic mice were randomly divided into five groups (eight mice per group) matched by body weight. Normal mice were administered with distilled water.

Forty-eight mice were divided into eight groups of six animals and had free access to water and food, and then animals were treated for five weeks as follows: Group 1: normal mice, Group 2: diabetic control mice, Group 3: Dihydrochalcone (25 mg/kg/day p.o) treated diabetic mice, Group 4: Dihydrochalcone (50 mg/kg/day p.o) treated diabetic mice, Group 5: Dihydrochalcone (100 mg/kg/day p.o) treated diabetic mice, Group 6: Mentformin (AG) (200 mg/kg/day p.o) treated diabetic mice as a standard drug.

### 4.5. Preparation of Urine, Serum, and Kidney Homogenate

After 5 weeks of treatment, diabetes mice were placed inside individual metabolic cages for 24 h and urine was collected. Body weight, kidney weight, food, and water intake were recorded; after 12 h fasting, the blood samples were collected from the tail vein. The animals were then sacrificed by asphyxiation using carbon dioxide. The serum was centrifuged for 20 min at 877× *g* and stored at −80 °C for further assays. After the kidneys were removed, weighed and washed with phosphate buffer saline (PBS pH 7.4). The right kidneys were homogenized in ice-cold PBS, and stored in liquid nitrogen for further biochemical and molecular assays. The left kidneys were removed and used for histopathological examination.

### 4.6. Measured of Biochemical Parameters in Diabetic Mice

The plasma glucose concentration was determined using an enzymatic colorimetric method using a commercial kit (Sigma Aldrich, San Luis, MO, USA). Total urinary protein levels were measured using a Rat Urinary Protein Assay Kit (Chondrex, Redmond, WA, USA). In serum, creatinine was evaluated using a QuantiChrom™ Protein Creatinine Ratio Assay Kit (BioAssay Systems, San Francisco, CA, USA), urea, uric acid and blood urea nitrogen test (BUN) were measured using Assay Kits (Abcam, Cambridge MA, USA) according to the manufacturer’s instructions. Kidneys were dissected and weighed (wet weight).

### 4.7. Glycosylated Hemoglobin

Blood samples were obtained by ocular venipuncture into tubes with EDTA and centrifuged at 800× *g* at 4 °C for 10 min to remove packed cells and plasma. Using two volumes of water packed cells were lysed. Then was added one volume of carbon tetrachloride to the hemolysate, and refrigerated at 4 °C overnight. Lysate was centrifuged at 27,000× *g* at 4 °C, for 30 min supernatant removed and used to analysis. Colorimetric estimation was measured according to the procedure of Parker et al. [23]. One mL of diluted hemolysate (10 mg of hemoglobin or fructose standard) was mixed with 1 mL of oxalic acid reagent. The mixture was incubated for 60 min at 124 °C in sealed tubes. After incubation, the mixture was allowed to cool to room temperature and added 1 mL of the trichloroacetic acid to each tube, mixed, and filtered through a 15 cm × 0.5 (1.d) glass column with a glass wool plug in the bottom. To 1.5 mL of the filtering add thiobarbituric acid reagent (0.05 M) and measured the absorbance at 443 nm of glycated hemoglobin.

### 4.8. AGEs Levels in Kidney

The kidneys were homogenized in 2 mL of 0.25 M sucrose, followed by centrifugation at 5 °C with 900× *g* and the pellet obtained was resuspended in 2 mL sucrose and centrifuged again then both supernatant obtained were mixed. The proteins were precipitated by adding trichloroacetic acid (TCA) in equal volume and centrifugated at 5 °C at 900× *g*, then protein pellet was added 1 mL methanol to eliminate the lipid fraction. Then, washing the insoluble protein using 10% cooled TCA and centrifuged: after the residue was solubilized in 1 mL of 1 N NaOH and the AGEs concentration was evaluated fluorometrically with an emission at 440 nm and excitation at 370 nm, and the results were indicated as relative fluorescence units (RFU)/mg protein [24].

### 4.9. AGEs Levels in Serum

Mouse plasma was centrifuged to collect supernatants. AGEs levels in plasma were used to determine Absolute Fluorescence (AF; 80 μL) and Advanced Glycation Index (AGI; 20, 40, 80 μL diluted in 1 mL of PBS) [25] using a Micro-plate Reader (Thermo Fisher Scientific, Voltam, MA, USA). AGE fluorescence was indicated at excitation wavelength of 350 nm and emission wavelengths of 450 nm in a spectrofluorometric detector (BIO-TEK, Synergy, Salt Lake City, UT, USA). AF was in arbitrary units (AU) with respect to protein concentrations. AGI was produced by the three points of dilution and expressed as the slope of the line.

### 4.10. Measure of the Expression of Intracellular Adhesion Molecule (ICAM)-1

After 5 weeks of treatment, frozen cortex was chopped on ice, then flushed through a disposable filter (100-mm; BioScientific Corp, Austin, TX, USA) with cold saline solution and collected. The solution was then flushed through filter (70-mm). The glomerular solution was obtained by inversion of the filter (70-mm), and afterward was flushed with a cold saline solution, followed by spinning at 6000 rpm at 4 °C for 15 min. The supernatant was eliminated and the pellets were suspended in 1 mL of Trizol. The presence of glomeruli was supported in a microscopy. Protein obtained from kidney cortex were used to evaluate the level of the expression of (ICAM)-1 (BioScientific Corp, Austin, TX, USA) according to ELISA kit instructions.

### 4.11. Histopathology

Kidney fragments were fixed in 10% neutral buffered formalin solution, dehydrated in ethanol, embedded in paraffin, and sectioned at 5 μm thickness using a rotary microtome. After dehydration, sections were stained with hematoxylin and eosin (HE). To evaluate the histopathological damage, each image of sections was examined for microscopic observations (400; Nikon, Tokyo, Japan).

### 4.12. Statistical Analyses

The data are presented as mean ± standard deviation. The significance of the differences was analyzed by one-way ANOVA with the Dunnett’s test using GraphPad Prism 7.0 for Windows (GraphPad Software Inc., San Diego, CA, USA). The value of statistical significance was established at *p* < 0.05.

## Figures and Tables

**Figure 1 molecules-24-01214-f001:**
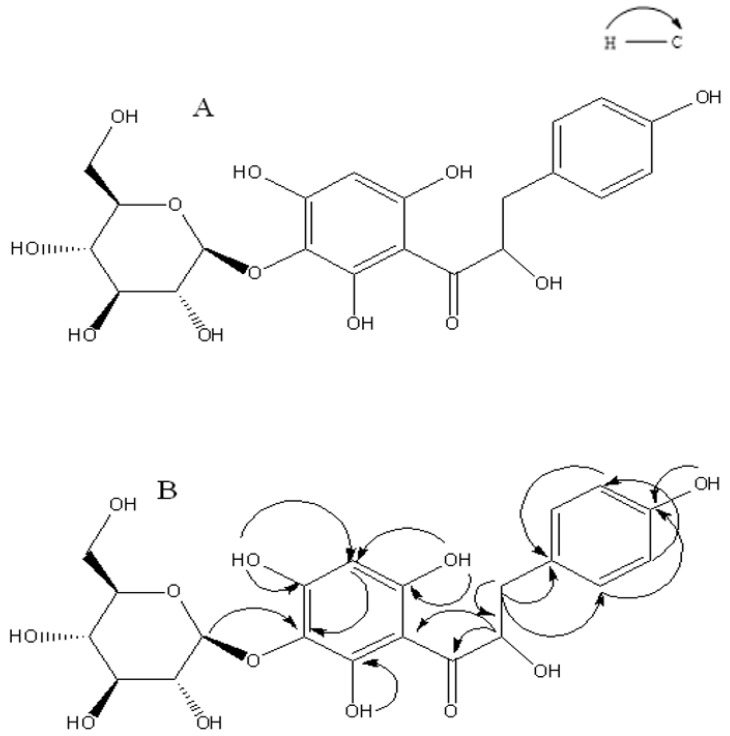
(**A**) Dihydrochalcone **1** isolated from bark of *Eysenhardtia polystachya*; (**B**) Heteronuclear Multiple Bond Connectivity (HMBC) of **1**.

**Figure 2 molecules-24-01214-f002:**
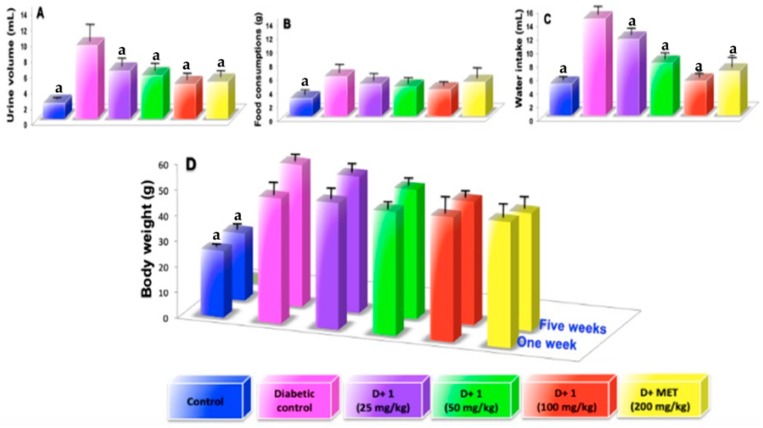
(**A**) examination of urine volume at the fifth weeks; (**B**) examination of food consumptions at the fifth weeks; (**C**) examination of water intake at the fifth weeks; (**D**) examination of body weigh at the fifth weeks; Results are expressed as mean ± SD; ^a^
*p* < 0.05 vs. diabetic control.

**Figure 3 molecules-24-01214-f003:**
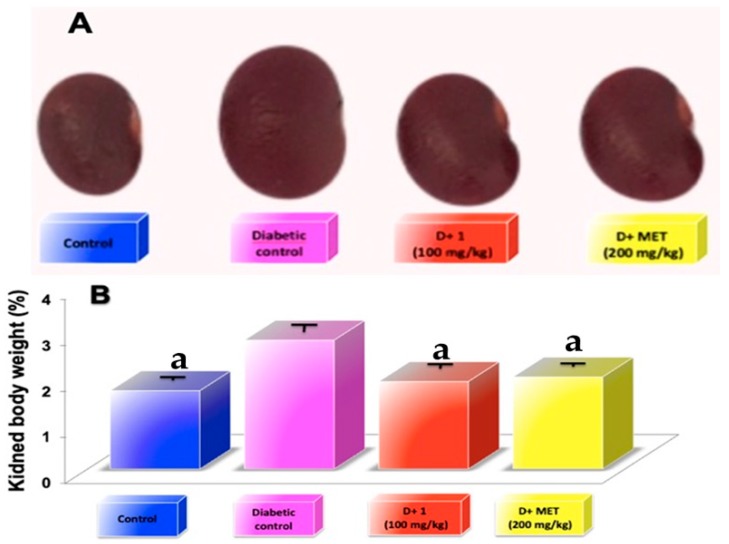
(**A**) Images of kidney representative of five weeks of the different treatments; (**B**) examination of kidney weight at the fifth weeks; Results are expressed as mean ± SD; ^a^
*p* < 0.05 vs. diabetic control.

**Figure 4 molecules-24-01214-f004:**
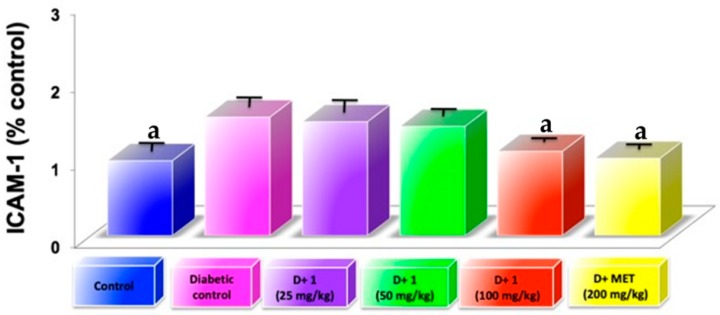
Examination of inflammation markers ICAM-1 in glomeruli of kidneys from diabetic mice at the fifth weeks; Results are expressed as mean ± SD; ^a^
*p* < 0.05 vs. diabetic control.

**Figure 5 molecules-24-01214-f005:**
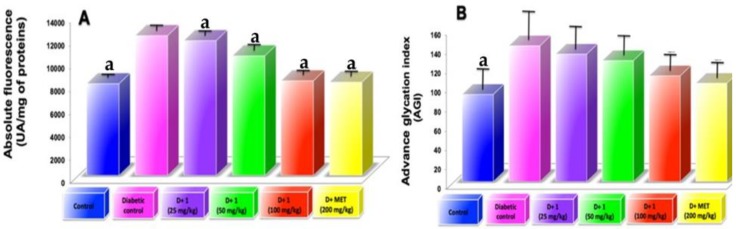
(**A**) Circulating AGE levels measured as absolute fluorescence at 5 weeks of experimental period; (**B**) Circulating AGE levels expressed as advanced glycation index. Results are expressed as mean ± SD; ^a^
*p* < 0.05 vs. diabetic control.

**Figure 6 molecules-24-01214-f006:**
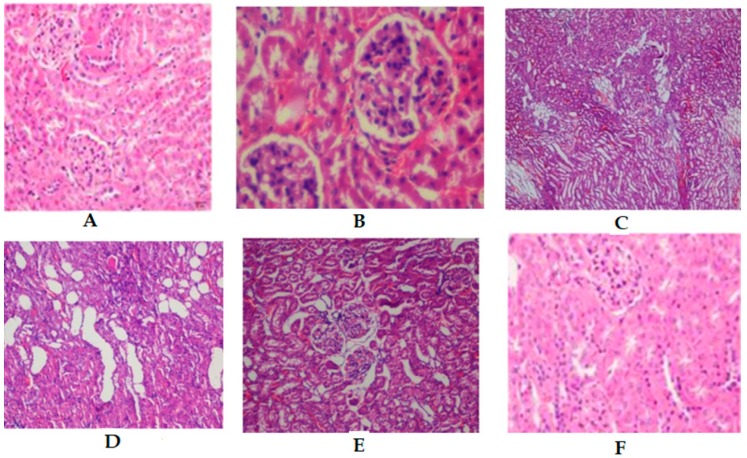
Kidney sections from mice by light microscopy (H and E stained). (**A**) Control group, (**B**) Diabetic control, (**C**) D + **1** (25 mg/kg), (**D**) D + **1** (50 mg/kg), (**E**) D + **1** (100 mg/kg), (**F**) D + MET (200 mg/kg).

**Table 1 molecules-24-01214-t001:** Effect of dihydrochalcone (**1**) on urine protein of diabetic-nephropathy mice.

Groups	Urine Protein (mg/dL)	
	0 Weeks	5 Weeks
NormaL ControL	0.45 ± 0.03 ^a^	0.44 ± 0.08 ^a^
Diabetic ControL	2.82 ± 0.7	3.61 ± 0.4
D + 1 (25 mg/dL)	2.81 ± 0.9	2.69 ± 0.5
D + 1 (50 mg/dL)	2.88 ± 0.6	2.16 ± 0.7
D + 1 (100 mg/dL)	2.89 ± 1.0	1.72 ± 0.9 ^a^
D + Met (200 mg/dL)	2.80 ± 1.3	1.70 ± 0.6 ^a^

Results are expressed as mean ± SD; ^a^
*p* < 0.05 vs. diabetic control (D).

**Table 2 molecules-24-01214-t002:** Effect of dihydrochalcone (**1**) on biochemical parameters in serum of diabetic-nephropathy mice.

Groups	BUN (mg/dL)		Creatinine (mg/dL)		Uric Acid (mg/dL)		Urea (mg/dL)	
	0 Weeks	5 Weeks	0 Weeks	5 Weeks	0 Weeks	5 Weeks	0 Weeks	5 Weeks
NormaL ControL	15.89 ± 3.4 ^a^	16.34 ± 2.8 ^a^	0.73 ± 0.03 ^a^	0.073 ± 0.03 ^a^	5.12 ± 1.4 ^a^	5.11 ± 1.0 ^a^	35.06 ± 4.3 ^a^	36.49 ± 1.0 ^a^
Diabetic ControL	36.12 ± 4.1	42.51 ± 2.9	3.01 ± 0.9	3.90 ± 0.7	13.11 ± 3.7	16.20 ± 3.1	80.10 ± 5.9	90.38 ± 6.4
D + 1 (25 mg/dL)	38.24 ± 2.9	30.47 ± 3.5 ^a^	3.25 ± 1.0	2.47 ± 0.5	13.30 ± 2.8	8.13 ± 2.5 ^a^	82.12 ± 3.7	69.28 ± 5.3 ^a^
D + 1 (50 mg/dL)	37.23 ± 5.3	25.48 ± 4.1 ^a^	3.94 ± 0.8	2.10 ± 0.4 ^a^	12.94 ± 1.7	6.24 ± 2.9 ^a^	83.43 ± 4.2	58.19 ± 6.1 ^a^
D + 1 (100 mg/dL)	38.48 ± 4.6	22.11 ± 3.9 ^a^	3.76 ± 0.6	1.70 ± 0.08 ^a^	13.84 ± 1.5	5.41 ± 1.6 ^a^	81.71 ± 5.3	50.13 ± 3.7 ^a^
D + Met (200 mg/dL)	38.40 ± 5.0	21.43 ± 4.2 ^a^	3.83 ± 0.9	1.55 ± 0.6 ^a^	12.67 ± 3.6	3.43 ± 2.3 ^a^	82.84 ± 3.8	56.22 ± 4.6 ^a^

Results are expressed as mean ± SD; ^a^
*p* < 0.05 vs. diabetic control (D).

**Table 3 molecules-24-01214-t003:** Effect of dihydrochalcone (**1**) on glycosylated haemoglobin levels and AGEs in Kidney of diabetic mice.

Groups	Glycosylated Haemoglobin (%)	AGEs (RFU/mg Protein)
NormaL controL	3.52 ± 0.7 ^a^	1.71 ± 0.9 ^a^
Diabetic controL	9.45 ± 1.1 ^a^	3.8 ± 0.8
D + 1 (25 mg/dL)	7.61 ± 1.8	3.0 ± 0.7
D + 1 (50 mg/dL)	6.27 ± 0.9 ^a^	2.6 ± 0.9
D + 1 (100 mg/dL)	4.3 ± 0.8 ^a^	2.0 ± 0.6
D + Met (200 mg/dL)	4.9 ± 0.6 ^a^	1.9 ± 0.7

Results are expressed as mean ± SD of five mice in each group to five weeks of experimentation. Values ^a^
*p* < 0.05 vs. diabetic control (D).

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
