# Peer review of "3′-O-β-d-glucopyranosyl-α,4,2′,4′,6′-pentahydroxy-dihydrochalcone, from Bark of Eysenhardtia polystachya Prevents Diabetic Nephropathy via Inhibiting Protein Glycation in STZ-Nicotinamide Induced Diabetic Mice"

_molecules, 2019, doi:10.3390/molecules24071214_

Round 1

Reviewer 1 Report

The objectives were investigated whether hydrochalcone 3΄-C-β24 glucopyranosyl-α,2΄,4΄,3,4-pentahydroxy-dihydroxy-chalcone (1), a powerful antiglycation compound previously isolated from Eysenhardtia polystachya, is able to protect against diabetic nephropathy in diabetic mice induced by Streptozotocin (STZ). The study suggests that hydrochalcone treatment protects renal function and avoids renal injury in STZ-nicotinamide-induced diabetic nephropathy, improved diabethic nephropathy markers as well as inflammation,  inhibition of advanced glycation end products in the kidney and circulation.The authors suggest that the renoprotective effect of hydrochalcone isolated from E. polystachya may be associated with blockade accumulation and formation of advanced glycation end products in the pathogenesis of diabetic nephropathy

Author Response

Rev 1

Comments and Suggestions for Authors

The objectives were investigated whether hydrochalcone 3΄-C-β-glucopyranosyl-α,2΄,4΄,3´,4-pentahydroxy-dihydroxy-chalcone (1), a powerful antiglycation compound previously isolated from Eysenhardtia polystachya,

It dihydrochalcone was previously isolated,  although in that occasion with similar spectra to the compound 3΄-O-β-glucopyranosyl-α, 2΄,4΄,6΄-trihydroxy-4-methoxy dihydrochalcone except for the substitution of an hydroxyl for the methoxyl group, we assumed by a mistake that it was a known compound. Actually when reviewing its spectral data again we find that it is a new dihydrochalcone derived from nothofagin. Therefore, the name (3΄-O-β-D-glucopyranosyl-α,4,2΄,4΄,6´-pentahydroxy-dihydrochalcone ) and structure  of dehydrochalcone were corrected

is able to protect against diabetic nephropathy in diabetic mice induced by Streptozotocin (STZ). The study suggests that hydrochalcone treatment protects renal function and avoids renal injury in STZ-nicotinamide-induced diabetic nephropathy, improved diabethic nephropathy markers as well as inflammation,  inhibition of advanced glycation end products in the kidney and circulation.The authors suggest that the renoprotective effect of hydrochalcone isolated from E. polystachya may be associated with blockade accumulation and formation of advanced glycation end products in the pathogenesis of diabetic nephropathy

Considering that AGE showed a closer relationship to oxidative damage and inflamation

and can best be described as chronic long-lasting damage. AGE increase  plasma  levels  are due to impaired renal clearence and elevated protein bound AGEs. Renal accumulation of AGEs has been linked to the progression of diabetes nephropathy.

This study demonstrated the utility of  using a cross-link breaker and an anti- protein oxidation such as dihydrochalcone (was previously determined)  could be a important treatment for diabetes nephropathy with effects on mediators of renal inhury.

Therapeutic effect of dihydrochalcone could be attributed in par to its ability to react with reactive carbonyl species and  cleavage of pre-formed AGEs within the kidney by a cross-link breaker inhibiting AGEs-formation.

We conclude that  1 mediates the antidiabetic activity mainly via glycation  inhibition, improved renal function.

Reviewer 2 Report

Although this manuscript provides some interesting scientific results several deficiencies should be addressed before acceptance for publication in the Molecules.

Although it is described as reported in reference 10, compound 1 is not described in this reference. Only this analogous compound is described. For new compounds, spectral data, detailed structure determination, absolute configuration of hydroxyl groups should be described.

Appropriate statistical methods should be used instead of post-hoc Duncan procedures.

Do you study compound 1 or hydrochalcone (CAS Registry Number 1083 - 30 - 3)? The description should be clarified.

Please describe the structural formula accurately (especially glucose).

Please describe the purity of the administered compound.

Is administration of Compound 1 a single dose or a continuous administration for 5 weeks? Please describe dosing schedule in detail.

Please describe the number of mice used in the experiment.

Author Response

Comments and Suggestions for Authors

Although this manuscript provides some interesting scientific results several deficiencies should be addressed before acceptance for publication in the Molecules.

 Although it is described as reported in reference 10, compound 1 is not described in this reference. Only this analogous compound is described. For new compounds, spectral data, detailed structure determination, absolute configuration of hydroxyl groups should be described.

It dihydrochalcone was previously isolated,  although in that occasion with similar spectra to the compound 3΄-O-β-glucopyranosyl-α, 2΄,4΄,6΄-trihydroxy-4-methoxy dihydrochalcone except for the substitution of an hydroxyl for the methoxyl group, we assumed by a mistake that it was a known compound. Actually when reviewing its spectral data again we find that it is a new dihydrochalcone derived from nothofagin. Therefore, the name (3΄-O-β-D-glucopyranosyl-α,4,2΄,4΄,6´-pentahydroxy-dihydrochalcone ) and structure  of dehydrochalcone were corrected

The characterization of the structure was described extensively

Appropriate statistical methods should be used instead of post-hoc Duncan procedures.

The statistical analysis was done using the Dunnett test, we apologize for having put incorrect information when reporting that the Duncan trial had been used.

Do you study compound 1 or hydrochalcone (CAS Registry Number 1083 - 30 - 3)? The description should be clarified.

The characterization of the structure was described extensively and the mistake in the name and structure was corrected.  As it is a new molecule it does not have CAS Registry Number

Please describe the structural formula accurately (especially glucose).

The  structural formula was improved

Please describe the purity of the administered compound.

The purity of the compound was checked using TLC with numerous mixtures of eluents, obtaining in each case a single spot.

 Is administration of Compound 1 a single dose or a continuous administration for 5 weeks? Please describe dosing schedule in detail.

 Schedule is described in detail.

Please describe the number of mice used in the experiment.

The number of animals used in the experiment was included

Round 2

Reviewer 2 Report

The modified figure is not attached.

TLC alone can not prove to be a single compound. That is insufficient data.

Please unify the name of the compound. Formally, compound 1 is not dihydrochalcone, but should be 3 - O - β - D - glucopyranosyl - α, 4,2,4 ', 6' - pentahydroxy - dihydrochalcone.

Is the absolute stereostructure of compound 1 R or S?
